# Effects of Air Pollution on Lung Innate Lymphoid Cells: Review of In Vitro and In Vivo Experimental Studies

**DOI:** 10.3390/ijerph16132347

**Published:** 2019-07-02

**Authors:** Bertha Estrella, Elena N. Naumova, Magda Cepeda, Trudy Voortman, Peter D. Katsikis, Hemmo A. Drexhage

**Affiliations:** 1Facultad de Ciencias Médicas, Universidad Central del Ecuador, Quito 170521, Ecuador; 2Friedman School of Nutrition Science and Policy, Tufts University, Boston, MA 02111, USA; 3Department of Epidemiology, Erasmus Medical Center, 3015GD Rotterdam, The Netherlands; 4Department of Immunology, Erasmus Medical Center, 3015GD Rotterdam, The Netherlands

**Keywords:** lung innate lymphoid cells, ILC, air pollutants, airway hyperresponsiveness

## Abstract

Outdoor air pollution is associated with respiratory infections and allergies, yet the role of innate lymphoid cells (ILCs) in pathogen containment and airway hyperresponsiveness relevant to effects of air pollutants on ILCs is poorly understood. We conducted a systematic review to evaluate the available evidence on the effect of outdoor air pollutants on the lung type 1 (ILC1) and type 2 ILCs (ILC2) subsets. We searched five electronic databases (up to Dec 2018) for studies on the effect of carbon monoxide (CO), sulfur dioxide (SO_2_), nitrogen dioxide (NO_2_), diesel exhaust particles (DEP), ozone (O_3_), and particulate matter (PM) on respiratory ILCs. Of 2209 identified citations, 22 full-text papers were assessed for eligibility, and 12 articles describing experimental studies performed in murine strains (9) and on human blood cells (3) were finally selected. Overall, these studies showed that exposure to PM, DEP, and high doses of O_3_ resulted in a reduction of interferon gamma (IFN-γ) production and cytotoxicity of ILC1. These pollutants and carbon nanotubes stimulate lung ILC2s, produce high levels of interleukin (IL)-5 and IL-13, and induce airway hyperresponsiveness. These findings highlight potential mechanisms by which human ILCs react to air pollution that increase the susceptibility to infections and allergies.

## 1. Introduction

Air pollution exposure is associated with an array of respiratory problems, particularly in children because their lungs and immune system are still maturing [1,2,3,4]. Development of lung infections and exacerbation of allergic airway diseases are more frequently found in people living in highly polluted areas [5,6,7,8]. In fact, we found that Ecuadorian children highly exposed to CO, NO_2_, fine particulate matter with diameters no greater than 2.5µm (PM_2.5_) or volcanic ashes presented increasedsusceptibility to respiratory infections [7], high rates of hospitalization for pneumonia [8], and elevated rate of emergency room visits due to acute upper and lower respiratory infections and asthma-related conditions [9]. 

PM, DEP, O_3_, and other chemicals and polluting compounds have been shown to have deleterious effects on the respiratory function of humans [3,5,6,7,8,10,11,12,13]. Exposure to these outdoor environmental pollutants can result in acute airway inflammation [14,15], increased mucosal secretions [16], oxidative lung damage [17,18,19], and loss of antibacterial functions [20,21,22,23,24]. These conditions may be mediated by a harmful effect of the pollutants on lung immune cells.

The effects of specific air pollutants on certain *classic* lung innate immune cells including alveolar macrophages (AM), polymorphonuclear (PMN), and dendritic cells (DC) have been recently gaining the attention [25,26,27,28]. For instance, it has been reported that particulate matter alters the anti-mycobacterial function of human respiratory epithelium [29] and that DEP impairs antibacterial immunity by suppressing the nucleolar factor NF-kβ pathway in human blood monocytes [30]. Also, concentrated urban particles have been shown to hamper bacterial clearance by AMs and PMNs in mice [31], whereas DEP induces activation of inflammatory signaling molecules and cytokine synthesis in AMs [32]. However, not much is known about the effects of air pollutants on newer sets of immune cells, so-called innate lymphoid cells (ILCs), which play an essential role in lymphoid tissue formation, tissue remodeling, tissue homeostasis [33,34], inflammation, and regulation of host responses to infection [34,35]. These cells are involved in the initiation, modulation, and resolution of lung diseases [36].

ILCs are derived from a bone marrow common lymphoid progenitor [37], have a lymphoid morphology, and lack both antigen-specific receptors and myeloid phenotypic markers. They populate barrier surfaces, including skin, intestine, lung, and some mucosal-associated lymphoid tissues [36]. Three major groups of ILCs have been defined based on the transcription factors needed for their development and the cytokines they produce [37,38,39]. Figure 1 summarizes the current understanding of ILC types, functions and activation pathways. The ILC1 group includes classical natural killer (NK) and non-NK cells, which are cytotoxic and produce interferon gamma (IFN-γ) [40,41]. The ILC2 group expresses interleukin (IL) IL-5 and IL-13 [41]. The type 3 (ILC3) group comprises of LTi cells, NKp46^−^ ILC3, and NKp46^+^ ILC3, which produce IL-17 and/or IL-22 [41]. Due to their cytokine production, ILCs 1, 2, and 3 resemble the adaptive T helper (Th) 1, Th2, and Th17 cells respectively [42]. In addition, human ILCs are highly heterogeneous among patients, tissues, and health conditions because they exhibit diversity in the expression of their surface markers [43,44].

In this paper, we systematically appraised all available studies to date that examined the effect of PM, O_3_, and DEP on lung ILC subsets in in vivo and in vitro models. While the role of ILCs in inflammation and the effects of air pollutants on classic innate immune cells are recognized, the effects of air pollutants on the lung ILCs remains scarce. Defining such relationships may help to understand how air pollutants affect respiratory infections and allergies. These investigations may also provide insights on potential strategies for improving diagnosis and treatment for these diseases.

## 2. Materials and Methods 

The systematic review was conducted using the general principles of the PRISMA-P (Preferred Reporting Items for Systematic Reviews and Meta-Analyses) [45] aimed to facilitate the development and reporting of systematic review protocols. The study protocol is available online [46]. 

### 2.1. Search Strategy

For the literature search, no limits were set on study designs, language or year of publication. We searched five electronic databases: Embase (via Embase.com), Medline (via Ovid), Web of Science, Cochrane Central, and Google Scholar from their inception until May 2018 for studies that measured the effect of air pollution exposure on number, viability, or function of any of the ILCs. The search strategy was constructed in cooperation with a medical information specialist (Wichor M Bramer, Medical Library, Erasmus MC) and combined terms related to air pollution (e.g., “air pollution”, “CO”, “DEP”, “PM2.5”, or “O3”) with those related to ILCs (e.g., “innate lymphoid cells”, “ILC1”, “ILC2”, “ILC3”, or “NK”), respiratory health (e.g., “asthma”, “respiratory infections”), or immunity (e.g., “innate immunity” or “cytokines”). We additionally checked references cited in selected articles and hand searched the PubMed engine database to retrieve additional articles. The full search strategy is provided in the Appendix A: Search terms, and Figure 2. 

### 2.2. Screening and Eligibility Criteria

Article titles and abstracts were initially screened for eligibility by two authors (B.E., M.C.). We included all studies based on cohorts, cross-sectional, experimental, in vitro, or in vivo designs in either animals or humans, that examined the effect of air pollutants exposure (measured by either pollution measurements or serologic markers of exposure) on lung ILCs and/or cytokines. We excluded systematic reviews, comments, consensus reports, editorials, guidelines, and protocols. 

### 2.3. Study Selection

First, we read the abstract of the retrieved references and selected those that included any of the specified respiratory health (e.g., asthma, airway hyperreactivity, and respiratory infection), as well as ILCs cell number and viability, cytokine production and activity, infection susceptibility and allergen-induced response, and airway hyperresponsiveness. Second, we read the full paper of the selected abstracts to confirm if they fulfilled all selection criteria. Disagreements were resolved by consensus and in consultation with a third independent reviewer (Josje Schoufour, Department of Epidemiology, Erasmus MC). 

### 2.4. Data Extraction

Extracted data from each article were registered in a predesigned form to record study design, analysis unit, and type of exposure, route and doses of exposure, experimentation arm/groups, findings, and conclusions (Appendix A: Methodological details of the studies). 

### 2.5. Quality Assessment of The Evidence

The quality of each study was assessed in terms of reproducibility of experimental methods and results, using a modified version of the Animal Research Reporting in vivo Experiments (ARRIVE) guidelines [47], and an adapted scale for in vitro experiments in human cells (Appendix A: Modified ARRIVE guidelines, and Appendix A: Adapted scale from ARRIVE guidelines for experimental studies in human cells).

### 2.6. Synthesis of the Evidence

The diversity in study designs, models, doses, and ways of assessing exposure and outcomes, did not allow us to carry out comparative quantitative analysis. Instead, we provided a qualitative overview of the extracted data and characterized the studies, exposures, outcomes, and the main findings.

## 3. Results

The search of electronic databases and hand searching provided 2209 citations. After removing duplicates, 1521 unique titles remained. Of these, 1499 studies were excluded based on the initial screening criteria. For the remaining 22 references, full-text papers were retrieved and further assessed for eligibility. From these, ten studies were not considered for the purpose of this review because one study did not measure lung ILCs and nine did not include the inorganic pollutants as the study objective. These studies were designed to induce allergic airway inflammation and observe how ILCs interact with the adaptive immune system, particularly Th2. (Appendix A) (For flow diagram see Figure 2). Out of the 12 remaining articles selected for the final evaluation, six studies investigated the effects of different pollutants on NK cells and six studies examined the effects on various other ILCs (Table 1). There were nine studies based on in vivo and in vitro murine experimental models [48,49,50,51,52,53,54,55,56], and three studies based on ex vivo and in vitro human cell models [57,58,59]. 

Regarding the quality of the experimental procedures, experimental animals, and experimental outcomes, all murine model studies met at least 75% of requirements stipulated by the modified ARRIVE guidelines. Experimental procedures and experimental outcomes of the human cell studies met at least 89% of requirements stipulated by the adapted scale (Appendix A: Modified ARRIVE guidelines, and Appendix A: Adapted scale from ARRIVE guidelines for experimental studies in human cells). These percentages ensure the reliability of the in vitro experimental procedures. 

Table 1 lists the 12 selected studies and their main findings, and Table 2 integrates the main findings per cell type, focusing first on ILC1-NK cells and then summarizing the effects of air pollutants on ILC2 cells. Overall, the studies show the diversity in study designs, models, doses, and ways of assessing exposure and outcomes. They were: (a)3 studies on the effects of O_3_ on ILC2 (all in mice) [51,52,55],(b)3 studies on the effects of O_3_ on NK cells (1 in mice and 2 in humans) [54,58,59],(c)1 study on the effects of carbon nanotubes on ILC2 (in mice) [48],(d)1 study on the effects of DEPs on ILC2 (in mice) [49],(e)2 studies on the effects of DEPs on NK cells (1 in mice and 1 in humans) [50,57],(f)1 study on the effects of PM_2.5_ on ILC2 (in mice) [56],(g)1 study on the effects of PM_2.5_ on NK cells (in rats) [53].

### 3.1. Effects of Air Pollutants on ILC1-NK Cells 

NK cells play a fundamental role in the immunity against intracellular infections and tumor immune surveillance (Figure 1). Overall, the summary showed that exposure to air pollutants results in decreased or impaired NK cell number, and modified cell function and cytokine release (Table 2).

#### 3.1.1. Cell Number and Viability

Studies in murine strains demonstrated that PM_2.5_ exposure decreases significantly both the absolute NK cell number in broncho-alveolar lavage fluid (BALF) and the NK cell influx into the airway lumen at 24 h post *S.*
*aureus* infection [53]. Similarly, DEP exposure decreased the number of NK cells in spleen in mice [50]. Concerning O_3_ exposure, this pollutant decreases the percentage of lung lymphocytes in mice, although without effect on their viability [54]. In contrast, in human cells, 3–5 days of low dose (1mg/mL) O_3_ exposure significantly increased the total NK cell population defined as CD3-CD16+/56+ without significant changes among the expression levels of other surface molecules [58]. 

#### 3.1.2. Cytokine Production

NK cells stimulated by IL-12 or IL-18, secreted from dendritic cells and macrophages, produce several cytokines, principally IFN-γ, IL-1β, IL-8, IL-17A, and tumor necrosis factor alpha (TNF-α), which are involved in lung inflammatory processes and infection resistance (Figure 2). DEP exposure caused strong, rapid in onset, long-lasting, and dose-related suppression of IFN-γ production in murine NK following lipopolysaccharide (LPS) stimulation; this suppression was due to the inhibition of the IL-12, and IL-18 response to LPS by accessory cells as well as by a direct inhibitory effect on IFN-γ mRNA levels, partly through post-transcriptional mechanisms [50]. In NK cells from healthy non-smoking non-asthmatic volunteers, DEP produced a modest increase of IL-1β, IL-8, and TNF-α release with no changes in IFN- γ [57]. O_3_ exposure also reduces the expression of IFN-γ on human NK cells by affecting the direct cell-cell interactions between epithelial and NK cells, and it is dependent on UL16 binding protein 3 (ULBP3) and major histocompatibility complex class I chain-related protein A and B (MICA/B) on epithelial cells [59]. 

#### 3.1.3. Activity

Cytotoxic activity of the NK cell is achieved through the release of granzyme B and perforin once the activating receptors have recognized their ligands (Figure 1). Activating receptors include NKp46, NKp44, CD16, CD69, and NKG2D. The murine experiments with the preceding PM_2.5_ exposures have demonstrated that: (a) exposure triggers a significant increase in bacterial load in the lung of rats infected by *S. aureus*; and (b) adoptive NK cell transfer to the lung of those rats markedly reduces the bacterial load to a level comparable to control rats that were infected with *S. aureus*, but not exposed to PM_2.5_. A potential mechanism explaining these observations is that alveolar macrophages, cultured with NK cells, have a high rate of phagocytosis of *S. aureus*, and suggests that interactions between NK cells and lung macrophages facilitate better control of bacterial infection by innate phagocytic cells [53]. In vitro and ex vivo experiments in human blood cells showed that exposure to DEP alone significantly reduces the cytotoxic potential of NK cells as compared to controls. At the same time, exposure to DEP, in the context of stimulation with the viral mimetic polyI:C, decreases the expression of CD16, granzyme B, and perforin, and suppresses the ability of NK cells to kill target cells without affecting the percent of NKG2D^+^ and NKp46^+^ cells [57]. In murine strains, continuous exposure to 1.0 ppm O_3_ for 1, 5, and 7 days had a significant immunosuppressive effect on pulmonary NK cell activity compared to controls, but this effect was reversed after 10 days of continuous O_3_ inhalation [54]. Furthermore, it has been demonstrated that the effect on pulmonary NK cells involved several cell types and/or their products that stimulate NK cells [54]. 

Similarly, studies on the interrelation between O_3_-exposed human epithelial cells and NK cells showed that direct exposure to O_3_ reduces, although non significantly, the expression of NK cell receptors (NKG2D and NKp46), the intracellular levels of granzyme B, and cytotoxicity function [59]. Yet, in another study, low doses of O_3_ exposure (1 mg/mL and 5 mg/mL) induced an increase in human NK-cell cytotoxicity without a significant difference between doses [58].

### 3.2. Effects of Air Pollutants on ILC2

ILC2 play critical roles in immune protection, tissue repair, brown fat biogenesis, and in the regulation of the inflammatory process (Figure 1). Air pollutants stimulate or inhibit ILC2 as shown in experiments in mouse models (Table 2).

#### 3.2.1. Cell Number and Viability

ILC2s (nuocytes) represent approximately 1% of the total number of cells in the whole lung lavage in mice [48]. Exposure to DEP plus house dust mite (HDM) increased the number of cytokine-expressing ILC2s along with Th2 T cells in the alveolar space of mice, but not in the lung tissue itself [49]. O_3_ exposure did not affect the total number of pulmonary ILC2 in non-obese mice [52,55], suggesting no influx or proliferation within 12 h after O_3_ exposure in such mice [52]. However, in obese mice O_3_ increased the number of IL-5^+^ and IL-13^+^ ILC2s [51]. Conversely, exposure to carbon in a multi-walled carbon nanotube experiment in two strains of mice resulted in a significant increase of ILC2 numbers [48]. 

#### 3.2.2. Cytokine Production and Activity

Exposure to DEP plus HDM also increased IL-5 and IL-13 levels in the BALF ILC2 compared with DEP alone and saline exposed groups, but this increase was also seen in Th2 cells, which was the principal source of those cytokines [49]. Experimental studies in mice showed that O_3_ [51,52] and multi-walled carbon nanotube [48] exposure induced the production of IL-33 by lung tissue which in turn activated several cells in the lung including Th2 T cells and ILC2 cells to produce IL-13 [48,51,52] and IL-5 [51,52]. In fact, it was reported that 12 h of O_3_ exposure significantly increased the transcription of IL-5 and IL-13 mRNA with the consequent increase of IL-5 and IL-13 production [52]. Additionally, it was confirmed that ILC2s were the principal source of IL-5 and IL-13 since O_3_ exposure of CD4^+^ Thy1^+^ Th cells isolated from the lungs did not induce those cytokines 12 h after exposure [52], and that bronchoalveolar lavage type 2 cytokines were not significantly reduced in O_3_-exposed obese mice when CD4+ T cells were depleted by treating mice with a depleting anti-CD4 antibody [51]. 

#### 3.2.3. Allergen-Induced Response and Airway Hyperresponsiveness (AHR)

IL-33, IL-5 and IL-13 are critical for AHR since they induce granulocyte infiltration and changes in airway epithelia (Figure 1). In an ovalbumin murine asthma model, exposure to PM_2.5_ exacerbated the asthma symptoms by significantly increasing the expression of RORα and GATA3 levels in peripheral blood mononuclear cell, transcription factors related to ILC2, suggesting that ILC2s play a crucial role in serious asthma induced by PM_2.5_ [56]. 

AHR was significantly enhanced by DEP exposure in HDM exposed mice due to accumulation of ILC2s and Th2 cells and type 2 cytokine production (IL5 and IL-13); however, the contribution of ILC2 after DEP exposure was marginal since ILC2 deficient mice exposed to DEP showed AHR depending on Th2 cells activation [49]. 

ILC-sufficient mice exposed for 9 days to ozone showed significantly greater BALF eosinophils, mucous cell metaplasia with more mucins in the proximal airway epithelium, and increased expression of lung mRNA transcripts associated with type 2 immunity than air-exposed mice [55]. Additionally, mice treated with anti-Thy1.2 antibodies, which significantly reduce a total number of both ILCs and T cells, dramatically lost the O_3_-induced influx of eosinophils and mucous cells metaplasia [55]. Importantly, intra-tracheal transfer of ILC2s to mice treated with anti-Thy1.2 showed that O_3_-exposed mice stimulated with methacholine present dramatically enhanced AHR compared to air-exposed mice [52]. 

Concerning carbon nanotube exposure, this regimen also induced AHR in mice through IL-13 action. The mechanism was via activation of epithelial cells to produce IL-33, which in turn recruits and stimulates ILC2 to produce IL-13 [48]. 

### 3.3. Effects of Air Pollutants on ILC3

No studies were identified that explored the effects of air pollutants on ILC3 numbers, viability, or cytokine production.

## 4. Discussion

In this systematic review, we showed that PM, DEP, O_3_, and carbon nanotubes affect ILCs in the respiratory system in two ways. The pollutants generally inhibit ILC1 (NK cell) cytotoxicity and cytokine (IFN-γ) production, thereby increasing the susceptibility to infections and allergies. The pollutants also stimulate ILC2 to produce IL-5 and IL-13, thereby increasing airway hyperresponsiveness. 

With regard to mechanism of action, the reviewed studies suggest that exposure to particulate contaminants (DEP and PM_2.5_) and high doses of O_3_ impair lung NK cell activity directly by decreasing expression of the activating receptors CD16 and CD69, granzyme B and perforin levels, and also indirectly by dampening the activity of neighboring cells implicated in NK cell activation. In addition, these same pollutants decrease IFN-γ secretion of NK cells, an important cytokine responsible for stimulating macrophages [60,61,62] and for inducing adaptive Th1 differentiation and function [63,64,65]. These alterations of NK cells jointly lessen the ability of the organism to fight against intracellular pathogens, viruses in particular, and may thereby result in increased susceptibility to respiratory infections. 

With regard to ILC2 cells, experimental studies in murine strains indicate that DEP, O_3_, and multi-wall carbon nanotubes activate ILC2 to produce IL-5 and IL-13, via a larger production of IL-33. As IL-5 and IL-13 are also produced by Th2 cells and are responsible for IgE class switching on B cells and the recruitment and activation of eosinophils [66,67,68,69], ILC2s stimulated by air pollutants likely act synergistically with Th2 cells to mount allergic respiratory processes, such as rhinitis and asthmatic episodes associated to air pollutants. 

This review contributes to our understanding of the role that ILCs play in the effect of environmental air pollutant exposure on respiratory infections and allergic disease. Our findings may be useful to identify potential intervention strategies, targeting the key molecules which enhance pulmonary NK cell function to protect against infections or to control an excessive activation of ILC2, thus decreasing the production of allergy-related cytokines and minimizing airway allergic inflammation. With respect to the development of preventive strategies, a comprehensive solution against harmful effects of air pollution on the respiratory health should require the implementation of and compliance with a broad range of policies to improve air quality [70]. 

We excluded nine studies that addressed the crosstalk between ILCs, particularly ILC2, and adaptive type 2 immune response during a natural airborne allergen-induced allergic airway inflammation, but not included any of the inorganic pollutants. The general conclusion is that both type of cells work together in a bidirectional way to maintain airway hyperreactivity. In the Appendix A we provided the list of the excluded studies.

This is the first comprehensive systematic review that builds a general picture of how ILCs are affected by different air pollutants and how respiratory health is affected. Since most of the evidence included in this review is based on in vitro and in vivo studies, thus the extrapolation of our findings to general population may be difficult. The experimental models could also underestimate or overestimate the effects of contaminants on the ILCs because a) the animal studies use inbred strains, which may differ in immune responsiveness to the heterogeneous responses found in the general population; b) the doses and duration of the exposure used in the experimental conditions may not represent real-life exposures; and c) the relationship between a dose of exposure and number of exposed cells may not accurately represent reality. Therefore, it would be important that future studies examine the effects of air pollution on human ILCs, for example in sputum samples and, preferably, in combination with reliable exposure measurements. These studies should help to understand the role and mechanisms underlying the participation of these cells in infections and airway hyperreactivity associated with air pollutants exposure. 

## 5. Conclusions

This systematic review of available studies on air pollution and ILCs shows that air pollution impairs the function of ILCs, increasing the susceptibility to infections and allergies. Exposure to key air pollutants, including PM, DEP, and O_3_ stimulate lung ILC2s to produce high levels of IL-5 and IL-13 and generally inhibit the cytotoxicity and IFN-γ production by ILC1-NK cells. These processes most likely play an essential role in the airway hyperresponsiveness and increased susceptibility to respiratory infections triggered by exposure to air pollution. Our findings also highlight substantial gaps in knowledge and the need to better understand the mechanisms by which human immune systems react to air pollution. 

## Figures and Tables

**Figure 1 ijerph-16-02347-f001:**
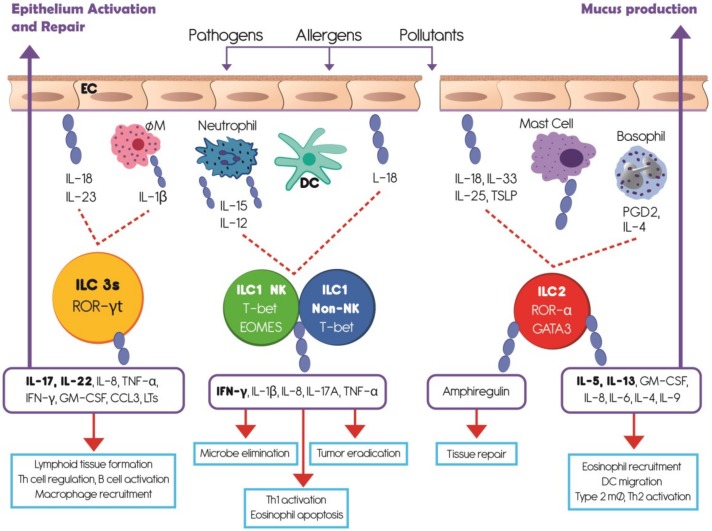
Summarizes the understanding of innate lymphoid cell (ILC) types, activation pathways and functions. Innate lymphoid cells are derived from a common lymphoid progenitor, have a lymphoid morphology, and lack antigen-specific receptors. Based upon the transcription factors needed for their development and the cytokines they produce, ILCs are divided in three groups which mainly populate barrier surfaces. ILC1 includes classical natural killer (NK) and non-NK cells and depends on the transcription factor T-bet. ILC2 depends on the transcription factors GATA3 and RORα. ILC3 requires the transcription factor ROR-γt and comprises a heterogeneous subset of cells. After external antigen contact, respiratory epithelial cells and classical innate immune cells produce several cytokines which stimulate different ILCs groups. IL-12, IL-15, IL-18 prime ILC1s to produce IFN-γ and other cytokines involved in microbe elimination, Th1 activation, and tumor eradication. ILC2s are activated by IL-4, prostaglandin D2 (PGD2), IL-33 and IL-25 to produce amphiregulin involved in tissue repair, IL-5 to recruit eosinophils, and IL-13 to stimulate mucus production by epithelial cells. ILC3s are primed by IL-18, IL-23, and IL-1β to produce principally IL-17 and IL-22 which participate in lymphoid tissue formation, Th cell regulation, B cell activation, and epithelium activation and repair. EC, epithelial cell; MØ, macrophage; DC, dendritic cell; PGD2, prostaglandin D2; TSLP, Thymic Stromal Lymphopoietin.

**Figure 2 ijerph-16-02347-f002:**
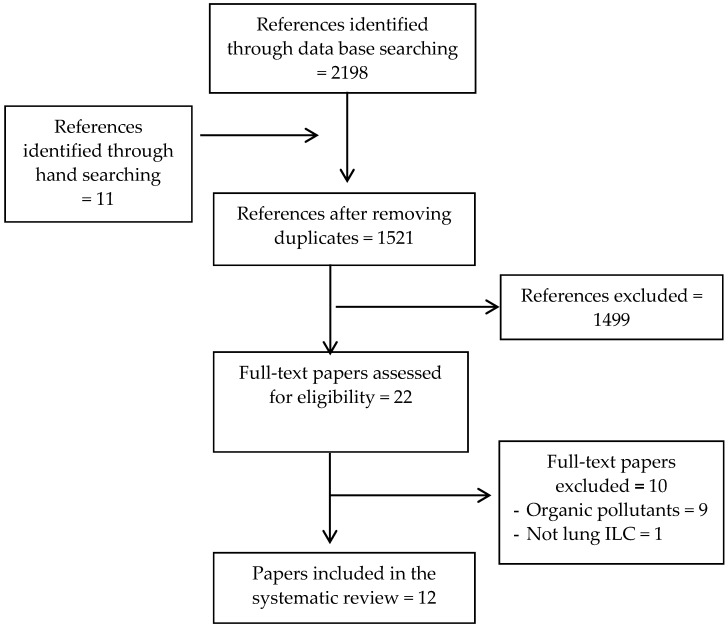
Flow diagram of study selection.

**Table 1 ijerph-16-02347-t001:** Characteristics and main findings of the studies.

Authors, Year	Type of Exposure (Doses; Method of Administration)	Outcome	Summary of Findings/Observed Effects of Exposure on the Outcome
***Type of Cell: Mice ILCs***			
Beamer, et al. 2013 [48]	Multi-walled carbon nanotubes (50 μg; oropharyngeal)	IL-33 function on ILC2	Epithelial cells (type II pneumocytes) in the lavage fluid induce secretion of IL-33 Elevated levels of IL-33 induce recruitment of ILCs in the airways ILCs acting in response to IL-33 stimulate AHR and eosinophil recruitment through the release of IL-13
***Type of Cell: Mice ILC2***			
De Grove, et al. 2016 [49]	DEP (25 mg on days 1, 8, and 15; intranasal)	Function and cytokine production	DEP alone has little effect but enhances the effects of house dust mite (HDM) exposure Marked increase in epithelium-derived cytokines IL-25 and IL-33 Increased numbers of DCs, neutrophils, ILC2s, CD41 T cells, CD81 T cells, and eosinophils. ILC2s marginally contribute to DEP-enhanced allergic airway inflammation Dysregulation of ILC2s and Th2 cells attenuated DEP-enhanced allergic airway inflammation. A crucial role for the adaptive immune system on concomitant DEP plus HDM exposure
Mathews, et al. 2017 [51]	O_3_ (2 ppm for 3 h; inhaled)	IL-33 action on ILC2 and γδ T	Interaction between Obesity and O_3_ Increased lung IL-13+ innate lymphoid cells type 2 (ILC2) and IL-13+ γδ T cells in obese mice Increased ST2+γδ T cells, indicating that these cells can be targets of IL-33, O_3_ induced type 2 cytokine expression in ILC2s and γδ T cells in obese mice Little or no effect of O_3_ on IL-33 in lean mice. ILC2s and γδ T appear to contribute to the effects of IL-33
Yang et al. 2016 [52]	O_3_ (3 ppm for 2 h on day 16; inhaled)	Il5 and Il13 RNA expression	O_3_ exposure increased airway levels of IL-33, a potent activator of lung ILC2s Lung-resident ILC2s were the predominant early source of the Th2 cytokines IL-5 and IL-13 in O_3_-exposed mice No ILC2 influx or proliferation within 12 h after O_3_ exposure ILC2s from the lungs: greater increased activation of Il5 and Il13 mRNA 12 h after O_3_
Kumagai, et al. 2017 [55]	O_3_ (0.8 ppm on day 1 or for 9 consecutive weekdays; inhaled)	ILC2 in airway inflammation, mucus cell metaplasia, and Type 2 immunity	O_3_ induced pulmonary esosinophilic inflammation in ILC sufficient mice O_3_ induced mucus cell metaplasia in proximal airway epithelium O_3_ increased mRNA transcripts of type 2 immunity in lung
Lu, et al. 2018 [56]	PM_2.5_ (25 mL/kg of a suspension of 15 g/L on days 1, 8, 15, and 21; intranasal)	ILC2-related transcription factors	Increased expression of RORα and GATA3 transcription factors, which are vital factor for ILC2. Increased IL33-levels which activates ILC2s
***Type of Cell: Rats NK***			
Burleson, et al. 1989 [54]	O_3_ (1.0 ppm for 23.5 h /day on 1, 3, 7, or 10 consecutive days; ambient)	Number and function of NK, and function of adherent cells	O_3_ induced suppression of pulmonary NK activity Cell/products involved in NK activation mediate the immunosuppression O_3_ decreased number but not viability of NK
Zhao, et al. 2014 [53]	PM_2.5_ (1, 5, or 10mg/kg body weight; intratracheal)	Number and bacterial response	PM_2.5_ increases susceptibility to respiratory infection by S. aureus. PM_2.5_ decreases the number of NK cells in the lung and suppress AM phagocytosis, which provides a potential mechanism to explain that association between ambient air pollution and pulmonary bacterial infections
***Type of Cell: Mice NK***			
Finkelman, et al. 2004 [50]	DEP (2 mg once; injected i.p.)	INF gamma production	DEP potently inhibits IFN-γ production by NK and NKT cells, which is rapid in onset, long lasting, and dose-related DEP induces an inhibitory effect on steady-state INF-γ mRNA levels and may also suppress INF-γ production through posttranscriptional mechanisms
***Type of Cell: Human NK***			
Müller, et al. 2013 [57]	DEP (10 μg/mL; direct exposure of cell)	Function and cytokine release	DEP reduced expression of the cytotoxic NK cell surface marker CD16, gene and protein expression of granzyme B and perforin, and the ability to kill target cells
Kucuksezer, et al. 2014 [58]	O_3_ (1, 5, 10, and 50 mg/mL cRPMI; direct exposure of cell)	Number, function	O_3_ increased number of CD16 cell and cytotoxicity of NK
Müller, et al. 2013 [59]	O_3_ (0.4 ppm; direct exposure of cell)	Effect of O_3_ exposed epithelial cells on natural killer cells function, cytokine release.	O_3_ reduced markers of activation, INF-γ production, and cytotoxic function. O_3_ upregulated ligands for NK in epithelial cells.

**Table 2 ijerph-16-02347-t002:** Effects of air pollutants on NK cells and ILC2: integration of findings.

		**NK Cell Features ^a^**				
**Exposure**	**PM _2.5_**	**DEP**		**O_3_**		**CN**
**Model**	**Rats**	**Mice**	**Human**	**Rats**	**Human**	**Mice**
Number	↓NK BALF ↓Influx into airways	↓NK in spleen		↓ % lung lymphocytes	Low doses: ↑ number	
Cytokine		↓IFN-γ	↑IL-1β ↑ IL-8 ↑ TNF-α no changes in INF- γ		↓ IFN-γ	
Activity	↑ Susceptibility to respiratory infection by *S. aureus*		↓ Cytotoxicity ↓ CD16 expression ↓ Granzyme B levels ↓ Perforin levels	↓ Pulmonary NK activity	↓ Cytotoxicity ↓ Granzyme B levels ↓ Markers of cytotoxicity Low doses: ↑ cytotoxicity	
**ILC2 cell features ^b^**						
**Exposure**	**PM _2.5_**	**DEP**		**O_3_**		**CN**
**Model**	**Rats**	**Mice**	**Human**	**Mice**	**Human**	**Mice**
Number		↑ in alveolar space but not in lungs		No effect in lean mice ↑IL-5+ and IL-13+ ILC2s in BAL in obese mice		↑ in lung
Cytokine		↑IL-5 ↑IL-13		↑IL-5 ↑IL-13		↑IL-13
Air way hiperresponsiveness (AHR)	Enhances AHR: ↑ RORα and GATA3 transcription factors related to ILC2	Enhances AHR: Accumulation of ILC2s and Th2 cells and type 2 cytokine production		Induces AHR: ↑ expression of lung mRNA transcripts associated with type 2 immunity		Induces AHR: ↑ IL-13 from ILC2

^a^ References: [50,53,54,57,58,59]. ^b^ References: [48,49,51,52,55,56]. CN = carbon nanotubes.

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
