# Peer review of "Effects of Air Pollution on Lung Innate Lymphoid Cells: Review of In Vitro and In Vivo Experimental Studies"

_ijerph, 2019, doi:10.3390/ijerph16132347_

Round 1

Reviewer 1 Report

As a reviewer I have the following remarks

Line 38: nitric dioxide (NO2), should be nitrogen dioxide.

Line 38: PM2.5 could be equal 2.5, better to say “no greater than 2.5”.

Line 109: “terms such as asthma, airway hyperreactivity, and/or respiratory infection..”, I am confused here by “and/or” – it is in the relation to what?, say: “asthma and respiratory infection”?

Thank you.

Reviewer 2 Report

Comments:

1.       Nine studies were excluded because the research focus was a mix of organic pollutants. Did any of these nine studies report data on ILCs? The authors should discuss why including these studies may not be appropriate for the purpose of this review. Adding a list of these excluded studies will help the readers to find them if interested.

2.       Nomenclature and grouping of ILCs was established very recently. Before this, ILCs were identified with various names. For example, ILCs that produce Th2-associated cytokines were called natural helper cells, nuocytes and innate helper 2 cells (Spits et al., 2013 – ‘Innate lymphoid cells – a proposal for uniform nomenclature’. Were such terminology considered while searching the databases for relevant research papers?

3.       Many studies involving ILCs use different set of surface and intracellular markers to identify these cells. If available, this information should be provided.

4.       Authors mention ‘ROR-gd’ as a transcription factor for ILC3s.  ‘ROR-gt’?

5.       The sentence in the legend for figure 2 ‘ILC3 requires the transcription factor RORgd and compromises some cells.’ is not clear. How and which cells do ILC3 compromise? This sentence should be rephrased.

6.       There are many typing errors throughout the manuscript. For example, the use of ‘house dust mice’ instead of ‘house dust mite’. These needs to be corrected.
